# Combining Virtual Surgical Planning and Patient-Specific 3D-Printing as a Solution to Complex Spinal Revision Surgery

**DOI:** 10.3390/jpm13010019

**Published:** 2022-12-22

**Authors:** David A. M. Tredan, Ralph J. Mobbs, Monish Maharaj, William C. H. Parr

**Affiliations:** 1NeuroSpine Surgery Research Group (NSURG), Sydney, NSW 2031, Australia; 2Neuro Spine Clinic, Prince of Wales Private Hospital, Randwick, NSW 2031, Australia; 3Faculty of Medicine, University of New South Wales (UNSW), Sydney, NSW 2052, Australia; 4Surgical and Orthopaedic Research Laboratories (SORL), Prince of Wales Clinical School, University of New South Wales, Randwick, NSW 2031, Australia; 53DMorphic Pty. Ltd., Matraville, NSW 2036, Australia

**Keywords:** 3D printing, complex spinal revision surgery, virtual surgical planning

## Abstract

With the advent of three-dimensional printing, rapid growth in the field and application in spinal and orthopedic surgery has been seen. This technology is now being applied in creating patient-specific implants, as it offers benefits over the generic alternative, with growing literature supporting this. This report details a unique application of virtual surgical planning and manufacture of a personalized implant in a case of cervical disc replacement failure with severe osteolysis and resultant hypermobility. Where this degree of degenerative bone loss would often necessitate a vertebrectomy to be performed, this case highlights the considerable customizability of 3D-printed patient-specific implants to contour to the bony defects, allowing for a smaller and safer operation, with the achievement of stability as early as 3 months after the procedure, by the presence of osseointegration. With increasing developments in virtual planning technology and 3D printing ability, the future of complex spinal revision surgery may adopt these technologies as it affords the patient a faster, safer, and less invasive and destructive procedure.

## 1. Introduction

Spinal fusion rates have shown a considerable increase over the last two decades, with estimates of the growth of up to 88% being recorded within the United States of America over a 16-year time interval [1]. It is no surprise that with this, we have seen significant technological advancements in interbody material selection and design [2]. Most recently, 3-Dimensional printing (3DP) technology has garnered much attention in the surgical field through its ability to manufacture patient-specific implant (PSI) options. Coupled with its wide field of application, it has allowed for the acceptance of 3DP technology into spinal surgery practice with significant enthusiasm. In essence, 3DP PSI manufacture relies on the successive addition of fine layers of specifically selected material to realize the precise 3D part [3,4]. Through this process, complex geometric designs can be realized with relative ease. The coupling of 3DP technology with Finite Element Model (FEM) analysis, a computerized process whereby predictions on how a prosthesis handles real-world forces, temperature, and other physical forces, has allowed surgeons to design prostheses that are not just highly customizable but also easily handle the intended mechanical forces being applied to them, this translating to lower chances of revision surgery and implant failure [5]. This was illustrated in a study by Mobbs et al. [6], where abnormal lumbar endplate topology necessitated the need for a 3DP PSI, with FEM analysis being an integral factor in the design phase of the implant, thus allowing the prosthesis to have a more uniform dispersion of axial loading forces over the entire endplate, this being superior to generic non-3DP options as analyzed via FEM analysis in this study [6]. The application of FEM is, however, not just reserved for the fabrication of 3DP PSI; this tool can also be used as an adjunct for surgeons, with an aim to minimize the risk of implant failure, by analyzing “generic” implants for tolerance of set physical forces, in the specific patient’s spinal anatomy, hence allowing for ideal implant selection [7,8]. This technology also has the ability to factor in the patient’s bone density and the density/material of the implant, further characterizing the potential for implant failure, with this technology being available in many other areas of orthopedic surgery [9,10]. It is no surprise that with the increasing awareness and accessibility of this technology, we are seeing a shift in the focus of clinical surgical practice towards a more personalized approach. With respect to 3DP technology, this trajectory is illustrated by surgeons’ increasing formation and use of patient-specific 3DP bio-models and guides, in addition to 3DP PSI, as evidenced by the growing body of medical literature [11]. In particular, a review article published by Vaishya et al. [12], describing the publication trends in 3D printing in orthopedics, demonstrated a yearly increase in publications surrounding 3D printing technology in orthopedic surgery [12].

Where generic non-customized implants mandate the distortion of the patient’s anatomy to fit the implant structure, 3DP PSI offers a precise tailor-made implant that negates the need for anatomy remodeling, thus potentially offering improved primary stabilization through an improved bone–implant interface leading to improved stress distribution, osteointegration, and the potential for a superior degree of spinal curvature correction [6]. These factors, in addition to improved speed of operation and less blood loss due to less invasive endplate preparation, may translate to improved surgical outcomes and, more importantly, better patient outcomes [13]. This is illustrated by Mobbs et al. [14], where the time for implant insertion of a 3DP PSI was reported as 90 s, while the insertion of the expandable “off the shelf” implant took over 40 min, with increased bleeding and requirement of more endplate preparation.

With increasing rates of spinal fusion, it is no surprise that the rates of spinal revision surgery, secondary to pseudoarthrosis, implant failure, and a lack of fusion or osteolysis, are also increasing [14,15,16]. What is concerning is that of patients who require revision spinal surgery, up to one-third may require multiple unplanned surgeries, with revision surgery often being more technically challenging with an associated increased risk of complication and need for a more extensive surgery [17,18]. It is here where the adoption of Virtual Surgical Planning (VSP) and 3DP PSI fabrication could garner considerable improvements in tackling complex revision spinal surgery cases over generic alternatives. 

PSI manufacture through 3DP technology is in its infancy. As such, the evidence surrounding it remains powered by case reports, often comprising complex deformity reconstruction and limited-size case series. Studies addressing cervical disc arthroplasty failure and its management, namely anterior cervical discectomy and fusion or vertebrectomy with or without posterior fixation, have been described. In particular, where severe osteolysis is present with cervical instability, common practice emphasizes using anterior vertebrectomy with vertebrectomy cage placement with the addition of posterior screw fixation [19]. It is here where the ability of 3D printing technology coupled with virtual surgical planning can realize 3DP patient-specific implants that contour to the highly irregular vertebral body topology (sequelae of the osteolysis), thus allowing for a smaller operation (interbody cage placement with integral screw fixation) to be performed, negating the need for a vertebrectomy and the associated increased risk to the patient that comes with this technique [20]. At the time of this publication, there was no study focusing on virtual surgical planning and manufacturing of a 3DP PSI in treating a failed cervical disc replacement with severe osteolysis and resultant hypermobility; this represents the first and novel use of 3DP PSI in the management of such pathology. The objective of this case report is to detail the integral use of Virtual Surgical Planning (VSP) in the production of a 3DP PSI in complex spinal revision surgery, with a focus on the ability of this technology to realize minimally invasive patient-specific prostheses options, minimizing the risk to the patient and allowing for smaller operation compared to conventional practice, for pathology with such severe osteolysis.

## 2. Materials and Methods

### 2.1. Case Presentation

A 39-year-old male presented with a one-year history of dynamic neck pain with associated crepitus. The patient was neurologically intact. His medical history included a C5/6 total disc replacement (M6-C cervical disc (Spinal Kinetics, Sunnyvale, CA, USA)), performed nine years prior for significant left C6 neuropathic pain. His initial procedure was without complication, and postoperative imaging revealed a well-placed disc replacement with an appropriate range of movement for the arthroplasty (Figure 1). Functionally, he had returned to full function within two months. Computed Tomography (CT) imaging revealed collapse (failure) of the cervical disc arthroplasty, with significant osteolysis of both the C5 and C6 vertebral bodies (Figure 1). Clinically, the patient had a hypermobile cervical spine, with worsening neck pain on flexion and extension.

Reviewing the CT DICOM images, the reconstruction options were limited to corpectomy with generic implant alternatives. With the assistance of the computer-assisted design engineer (W.C.H.P.), the neurosurgeon (R.J.M.) was able to determine the implant design. Virtual surgery planning (VSP) further provided implant design improvements allowing for the completion of the implant design. The implant was printed in Titanium alloy using an EOSM 100 3D printer (3DMorphic, Sydney, Australia) (Figure 2). The surgical procedure proceeded without complication and, as per the VSP, via a standard anterior cervical fusion exposure. 

### 2.2. Preoperative Planning

Upon receiving the CT DICOM images, preoperative planning included CT segmentation, 3D image reconstruction (Materialise MIMICS (version 22.0) and 3 Matic (version 14.0) Leuven, Belgium), VSP and 3DP of a biomodel, and finally, design of the PSI (3DMCAD, 3DMorphic, Sydney, Australia). The intraoperative goals focused on the safe removal of the collapsed implant, implantation of the new 3DP PSI with good bony purchase of the screws, and improvement of the sagittal balance of the patient’s cervical spine. The patient-specific implant was 3DP using direct metal laser solidification (DMLS) of biomedical grade 5 titanium alloy powder (Ti_6_Al_4_V). This metal alloy was selected for its relatively low stiffness, good mechanical properties, lightweight, corrosion resistance, and biocompatibility [21,22]. DMLS was chosen over other printing modalities due to its high precision and accuracy in manufacturing 3DP implants [23,24]. The prosthesis was printed (3DMorphic, Sydney, Australia) using an EOSM100 (Krailling, Germany) 3D printer. The implants were manufactured with three different heights and lordotic angles to improve intraoperative options for the neurosurgeon, optimizing the fit of the implant and translating to optimal anatomical reconstruction.

### 2.3. Operative Technique

The surgical technique performed was that of a standard anterior cervical surgical approach, which is well described. In brief, the patient was positioned supine, head neutral. Image-intensified fluoroscopy confirmed the C5/6 level before the incision. A horizontal incision was made to the right of the midline, and tissues were dissected until the prevertebral fascia was reached. The C5/6 level was confirmed by the presence of metalware, noting the presence of metallosis (Figure 3). Trimline and Caspar retractors were used to optimize exposure. The collapsed arthroplasty revealed no viscoelastic inner core, with rupture of the outer annulus of the initial device, and was removed piecemeal. Significant and diffuse osteolysis and metallosis were noted, and the latter was removed. After carefully selecting the PSI with the correct height, this device was implanted and affixed with two integral fixation screws (Matrix Medical innovations, Sydney, Australia) at a predetermined length based on the VSP. Bone grafting with custom allograft (Queensland Tissue Bank) was placed in the central aperture of the implant. Intraoperative X-ray confirmed the optimal placement of the PSI.

## 3. Results

Intraoperatively, the disc arthroplasty was found to have collapsed with the inner disc component destroyed (Figure 3). The fracture disc component, followed by the two titanium endplates, were removed with aid from curettes and pituitary rongeurs (Video S1). Metallosis was apparent and delicately debrided from the bony edges with the assistance of the curette and the high-velocity drill. After copious irrigation, the PSI was placed with excellent fit and primary stabilization. The surgical team noted that the combination of VSP, in particular its estimation of screw length and the 3DP biomodel, allowed for correct sizing of the PSI and the optimal screw length selection, improving both the speed of the operation, reduced surgeon stress, ease of workflow (Video S1) and minimizing the potential risks to the patient. The procedure was also reported to have less than 20 mL of blood loss.

The patient was extubated at the conclusion of the case without neurological deficits or complications. The day after the operation, the patient had nil mechanical neck pain. CT imaging revealed a well-seated PSI and improvement in the sagittal balance (Figure 4). The patient was discharged with instruction to wear an Aspen (semi-rigid) collar for two weeks. The patient returned to independence and sedentary employment at two weeks and full work duties three months after the operation. At three months follow-up, the patient remains asymptomatic, without neck pain, and working full time. Repeat CT imaging revealed ongoing appropriate seating of the implant and early osteointegration of the implant (Figure 4).

## 4. Discussion

The authors herein report the first application of VSP and 3DP PSI design and manufacture in treating cervical disc arthroplasty failure with severe osteolysis (complex spinal revision surgery). This technology affords the patient a smaller operation than the standard convention, as the technology’s customizability allows for the manufacture of a PSI whose endplates are married to the irregular topology of the patients’ endplates. This case, thus, identifies another application of this technology in spinal reconstruction and adds to the growing body of literature on the use of 3DP technology in complex spinal surgery.

On review of the surgical workflow of this case, VSP was instrumental in many ways, including distractor (e.g., Casper pin) placement, complex PSI design and placement based on the degree of bony osteolysis of the endplates, as well as in the pre-planning of screw length and trajectories. It was felt that the combination of VSP, 3DP biomodel, and 3DP PSI translated into a reduced requirement for “surgical improvisation” in order to facilitate optimal implant placement and stabilization (through screw placement). This was felt to lower the potential risk of complications such as prosthesis mispositioning, bony destruction, and bleeding. It was also noted that a significantly lower number of intra-operative radiographs were required until optimal prosthesis implantation.

A key consideration in spinal surgery with personalized implants is matching the prosthesis cage dimensions with that of the unique topology of the patient endplate morphology, with a view to maximizing the bone-to-prosthesis contact. This is beneficial as it allows for improved stress dispersion between the implant and the vertebral endplate, improved osteointegration, and in reducing prosthesis subsidence [25,26]. Through the adoption of personalized implant use, the difference in the device–endplate interface is minimized, translating to a decreased requirement for endplate preparation. Endplate preparation potentially weakens the vertebral body’s stiff endplates, predisposing to loading of the underlying cancellous bone, which may predispose to implant subsidence [26,27].

When facing complex spinal revision surgery secondary to prosthesis failure with severe osteolysis, it is common practice that the irregular bony surface is drilled to facilitate the placement of a generic “off the shelf” cage implant, often leading to the requirement of a more extensive exposure area and potential requirement of a 2-level procedure, to facilitate integral screw fixation placement [20,28]. As illustrated by this case, it is now possible to use VSP to segment the vertebral bodies with their severely irregular topology, despite the presence of the failed implant, and design a PSI that contours to the irregular surface, allowing the maintenance of the bony architecture, as well as correction in sagittal balance, at the pathological level. In addition, the planning ability of VSP in selecting appropriate screw length and trajectory allowed for accurate purchase in the remaining bone. Where this kind of procedure would involve much bony destruction, “surgical improvisation”, and trial and error of generic implants, the adoption of VSP and use of a 3DP PSI allowed for a simple, less destructive operation with the reassurance of correct screw placement and implant fit.

This case illustrates the possibilities that 3DP technology can bring to revision spinal surgery in patients with significant osteolysis, spinal instability, and associated implant failure. This case provides a strong argument for the application of VSP and 3DP PSI in complex revision spinal surgery. This is of particular value in cases where significant bony destruction has occurred due to the subsidence or loosening of implanted prostheses, where considerably altered bony topology can be encountered. This benefit is already being seen in cases where traumatic fractures are being addressed [29,30,31]. In addition, compared to traditional, highly destructive approaches for this pathology, the adoption of VSP with 3DP PSI means that patients could undergo smaller, less destructive surgeries with the same postoperative result, with a potentially smaller risk of complications. The limitation of this study is that it only comprises one case study, and as such, the level of evidence and impact it provides remains low. It is the hope that with future research in this area, that higher-powered studies may further iterate the same benefits that were seen in this case report, with the hope that this potentiates the regular utilization of 3DP technology in spinal revision surgery. 

## 5. Conclusions

As this case demonstrates, advances in VSP, PSI design, and 3DP manufacturing have allowed this technology to be applied in complex spinal revision surgery, where implant failure is the antecedent cause. In cases where significant bone destruction is present, as in this study, the current practice would usually mandate extensive reconstructive spinal surgery in the form of a vertebrectomy. 3DP technology can afford the patient a more minimalistic approach by providing a patient-specific implant that contours to the highly irregular endplate topology while providing the fusion and load-bearing capabilities required of the implant. This technology may herald a new era in the management of complex spinal reconstructive surgery, with a focus on personalized surgical healthcare; this being said, additional research in this area is required to further support the use of 3DP technology in this field.

## Figures and Tables

**Figure 1 jpm-13-00019-f001:**
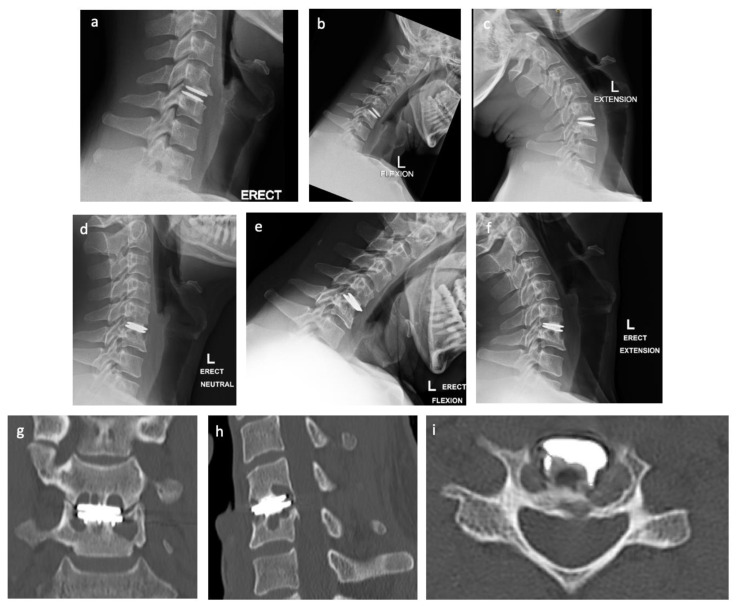
Postoperative radiographs 12 months after total disc arthroplasty via lateral view (**a**) and through a dynamic range of motion flexion (**b**) and extension (**c**). Postoperative radiographs nine years after total disc arthroplasty, via lateral view (**d**) and through a dynamic range of motion flexion (**e**) and extension (**f**), these images reveal arthroplasty collapse with dynamic instability. Postoperative Computed Tomography (CT) reveals the collapse of the disc arthroplasty and significant osteolysis in the coronal (**g**), sagittal (**h**), and axial (**i**) planes.

**Figure 2 jpm-13-00019-f002:**
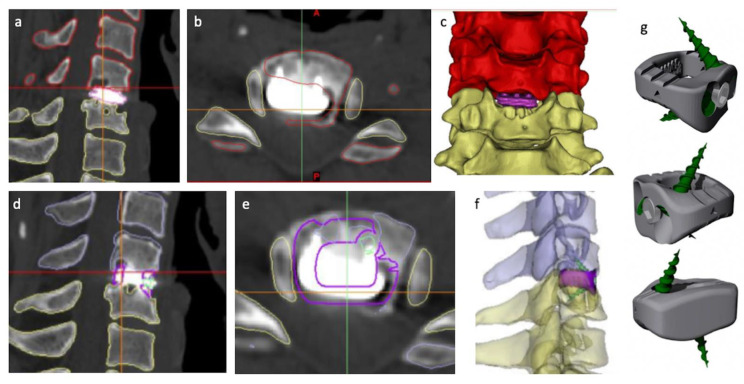
Virtual surgical planning. Preoperative CT imaging with superimposed segmentation borders of the C5 and C6 vertebral bodies, as well as the disc arthroplasty, viewed in the sagittal (**a**) and axial (**b**) planes with subsequent 3D reconstruction (**c**). Preoperative CT imaging with superimposed segmentation of the borders of the C5 and C6 vertebral bodies, as well as the proposed PSI design viewed in the sagittal (**d**) and axial (**e**) planes with subsequent 3D reconstruction with the addition of the proposed PSI design inlaid in the reconstruction (**f**). Note the deep purple line in images (**d**,**e**) represent the borders of the suggested PSI with particular note of the personalized contouring of the implant to the patients remaining bony morphology visualized in the sagittal view (**d**) as well as in the virtual 3D model of the implant (**g**). In the image (**f**), the planned trajectories and size of the integral fixation screws can be seen in green. The red and yellow lines in images (**a**,**b**), and blue and yellow lines in images (**d**,**e**), represent the cortical surfaces of the vertebrae.

**Figure 3 jpm-13-00019-f003:**
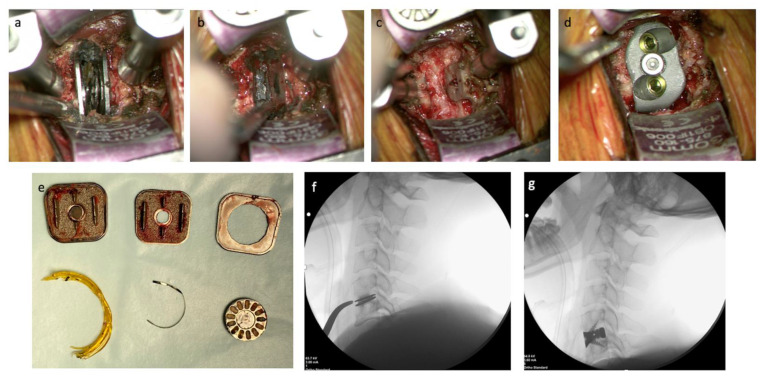
Intraoperative findings and prosthesis placement. Intraoperative image revealing a fractured cervical disc arthroplasty (**a**), with subsequent removal of the metalware, revealing the underlying metallosis (**b**). Intraoperative image displaying the significant osteolysis that had occurred to the vertebral endplates (**c**) and subsequent placement of the 3DP PSI and integral screw fixation (**d**). (**e**) An image of the broken parts of the M6-C cervical disc arthroplasty explanted from the patient. (**f**) An intraoperative radiograph reveals the collapsed disc replacement with evidence of osteolysis. (**g**) An intraoperative radiograph shows the appropriate placement of the new 3DP PSI with integral screw fixation.

**Figure 4 jpm-13-00019-f004:**
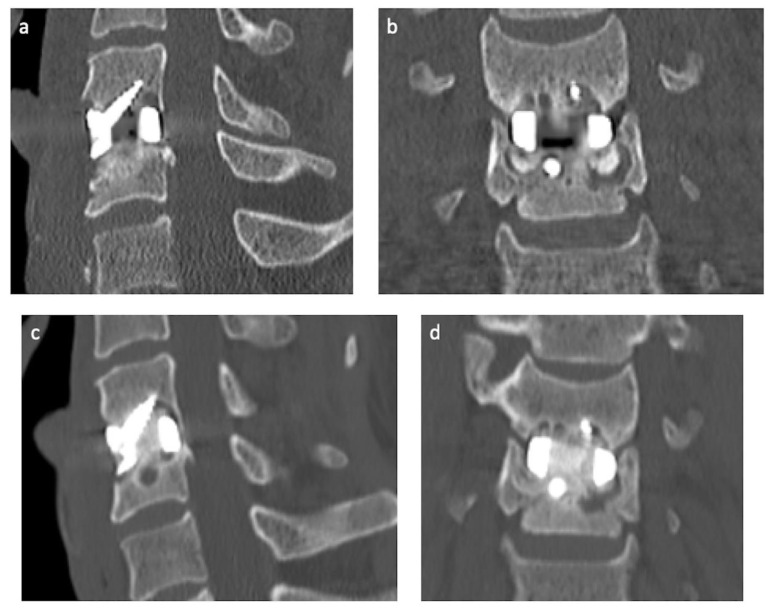
Postoperative CT imaging. Day-1 postoperative CT in the sagittal (**a**) and coronal (**b**) planes, revealed excellent prosthesis placement and bone purchase of the integral fixation screws. Three-month postoperative CT imaging in the sagittal (**c**) and coronal (**d**) planes, revealing ongoing excellent prosthesis positioning and evidence of osteointegration through the central aperture of the implant.

## Data Availability

No new data were created or analyzed in this study. Data sharing is not applicable to this article.

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
