# Peer review of "Combining Virtual Surgical Planning and Patient-Specific 3D-Printing as a Solution to Complex Spinal Revision Surgery"

_jpm, 2022, doi:10.3390/jpm13010019_

Round 1
Reviewer 1 Report
That is an interesting paper.
Author Response
Dear reviewer,
I appreciate the time you have taken reviewing this article. I am glad to hear that you have found it interesting. In response to your suggestion, I have reviewed the article and improved the grammatical fluency and small number of spelling error that you had most likely noticed. As some changes had to be made to the article a revised article will be uploaded. Thank you for your help.
Kind regards.
Reviewer 2 Report
1. The abstract section should be enhanced to include quantitative data.
2. Please give a “take-home” message as the conclusion of your abstract.
3. Rearrange keywords alphabetically.
4. The Reviewer do not see the novel in the present article. My examination revealed that several similar previous publications appear to appropriately address the issues you have brought up in the current submission. Please emphasize it more advance in the introduction section if there are any more truly something really new.
5. Previous studies must be explained in the introductory part, including their work, innovation, and limits, to demonstrate the research gaps that will be filled in the current study.
6. The end of a paragraph in the introduction section should explain the objective of the present article, the present form was not.
7. The authors need to explain some attempt to minimize revision surgery and implant failure, there are computational simulation via finite element method. It is a crucial issue that authors should provide in the introduction and/or discussion section. The MDPI's suggested reverence should be adopted as follows: Ammarullah, M. I.; Santoso, G.; Sugiharto, S.; Supriyono, T.; Wibowo, D. B.; Kurdi, O.; Tauviqirrahman, M.; Jamari, J. Minimizing Risk of Failure from Ceramic-on-Ceramic Total Hip Prosthesis by Selecting Ceramic Materials Based on Tresca Stress. Sustainability 2022, 14, 13413. https://doi.org/10.3390/su142013413
8. To let the reader, comprehend the workflow of the current study, the authors could include extra illustrations as a type of figure in the materials and methods rather than simply the main text as a present form.
9. Manufacturer, country, and specification information for experimental setup should be presented with more specificity.
10. The error and tolerance of the experimental tools employed in this investigation are critical details that must be explained in the publication. It would be a valuable discussion because of the differing outcomes in the subsequent study by other researchers.
11. Findings must be compared to similar past research.
12. Before moving on to the conclusion section, the present reports’ limitation must be added at end of the discussion section.
13. Conclusion section is missing, provide it.
14. Mention further research in the conclusion section.
15. Literature from the last five years should be enriched to reference. MDPI reference is strongly recommended.
16. In the whole of the manuscript, the authors sometimes made a paragraph only consisting of one or two sentences that made the explanation not clearly understood. The authors need to extend their explanation to become a more comprehensive paragraph. In one paragraph, it is recommended to consist of at least 3 sentences with 1 sentence as the main sentence and the other sentences as supporting sentences.
17. The manuscript needs to be proofread by the authors since it has grammatical and language issues.
18. After revision, provide a graphical abstract for submission.
Author Response
"Please see the attachment"
Response to Reviewer 2 Comments
Point 1: The abstract section should be enhanced to include quantitative data.
Response 1: This study is a case report describing a unique case of osseolysis secondary to cervical total disc replacement failure, and the novel use of 3D printing technology to print a patient specific implant, to facilitate stability in the patients cervical spine. As this is a case report, there is no large amount of quantitative data. Having said this, the presencce of fusion (osseointegration), a marker of successful fusion, was documented inthis patient. As such, this quantitative data was included in the revised absstract.
Point 2: Please give a “take-home” message as the conclusion of your abstract.
Response 2: Pin response to point 2, a take home message has been fortulated a the end of the abstract.
Point 3: Rearrange keywords alphabetically.
Response 3: Thankyou for highlighting this issue. It has beeen corrected.
Point 4: The Reviewer do not see the novel in the present article. My examination revealed that several similar previous publications appear to appropriately address the issues you have brought up in the current submission. Please emphasize it more advance in the introduction section if there are any more truly something really new.
Response 4: In response to point 4, the use of 3D printed patient specific implants for the correction of failed previous fusion prostheses has been reported in the literature before. Having said this, what makes this case unique is two-fold. Firstly, this case report documents the first revision of a failed cervical disc arthroplasty (total disc replacement prosthesis), through use of a 3D printed patient specific implant, where severe osseolysis has resulted in both cervical instability and a consideerably abnormal endplate topology. Secondly, in this case report, the patieent’s pathology was adressed through the employ of a 3D printed patient specific implant, which was designed to mimic an anterior cervical discectomy and fusion (ACDF) intervtebral cage. This is of importance and novel comparrd to current literature, as through the use of 3D Printing technology (at the virtual surgical planning stage, design stage and manufacturee stage) we were able to minimise the size of the implant and correspondingly minimising the extent of the surgergy, which by current standards would normally require tha the patient undergo a cervical verrtbrectomy, for the degree of bone osseolysis that was present. In summary this case report illustrates the highly customisable nature of this technology, and its ability to address pathology, which would norrmally (by current standards) require large scale reconstruction (vertebrectomy), through a smaller, safer and less destructive means.
A vast Pubmed search has again been conducted, with the key words: “3D print”, “additive manufacturing”, “implant”, “cervical arthroplasty”, “cervical disc replacement”, “patient specific” in order to see if articles had been published with a similar objective to ours. Again on reviewing the resultant titles and abstracts, there is still no study illustrating the key objective illustrated in our case report. It is thus the combined opinion of the authors that this case report illustrates a unique glimpse into possibilities that 3D printing technology, and the manufacture of patient specific implants, could hold in optimizing spinal revision surgery, through allowing for less invasive, safer and smaller operations.
Point 5: Previous studies must be explained in the introductory part, including their work, innovation, and limits, to demonstrate the research gaps that will be filled in the current study.
Response 5: In response to this point, and taking from information from the reponse from point 4, although there are studies that focus on the use of 3D printing to realise interbody implants with abnormal endplate topology “Mobbs, R. J.; Parr, W. C. H.; Choy, W. J.; McEvoy, A.; Walsh, W. R.; Phan, K. Anterior Lumbar Interbody Fusion Using a Personalized Approach: Is Custom the Future of Implants for Anterior Lumbar Interbody Fusion Surgery? World Neurosurg 2019. DOI: 10.1016/j.wneu.2018.12.144” (this reference being added and explained in the new revision of the article), therer are no studies in the literature which focus on designing an implant that contours to severely oosseolytic bone contours.
Point 6: The end of a paragraph in the introduction section should explain the objective of the present article, the present form was not.
Response 6: Thank you for highlighing this issue. A clear explanation as to the objective of the study, has been addeed to the revised manuscript, to correct this issue.
Point 7: The authors need to explain some attempt to minimize revision surgery and implant failure, there are computational simulation via finite element method. It is a crucial issue that authors should provide in the introduction and/or discussion section. The MDPI's suggested reverence should be adopted as follows: Ammarullah, M. I.; Santoso, G.; Sugiharto, S.; Supriyono, T.; Wibowo, D. B.; Kurdi, O.; Tauviqirrahman, M.; Jamari, J. Minimizing Risk of Failure from Ceramic-on-Ceramic Total Hip Prosthesis by Selecting Ceramic Materials Based on Tresca Stress. Sustainability 2022, 14, 13413. https://doi.org/10.3390/su142013413
Response 7: Thank you for highlighing this issue. This point has now been adressed in the article. I appreciate the reviewers suggestion as this point does improve the arrticle. All references have been reviewed and the style of referenceing listed in point 7 has now been employed in the article.
Point 8: To let the reader, comprehend the workflow of the current study, the authors could include extra illustrations as a type of figure in the materials and methods rather than simply the main text as a present form.
Response 8: In response to point 8, it is the opinion of the author that the figures in their current formation optimizes the readers understanding of the clinical pathology at play, the chronological steps in the operration and in the prosthesis design. The figures are currently located in the material and methods section and the result section.
Point 9: Manufacturer, country, and specification information for experimental setup should be presented with more specificity.
Response 9: In responsee to this point, the author has includded the manufacturerr, country and specification information of the implant (material used tto print, the specific printer details and the type of printing performed) into the rervised manuscript.
Point 10: The error and tolerance of the experimental tools employed in this investigation are critical details that must be explained in the publication. It would be a valuable discussion because of the differing outcomes in the subsequent study by other researchers.
Response 10: In response to point 10, a statement has been addde to the manuscript detailing the reason for the choice of direct metal laser sintering (DMLS) a the meethod of 3D printing, this beeing due to the precision in th eprinting process that can occur with this modalitty in coparison to others.
Point 11: Findings must be compared to similar past research.
Response 11: In response to this point, it is the combined opinion of the authors that there is not any previous research that would allow comparrison to our case report. As this, on our review of the literature, is the first case repot where 3D printing technology has been employed to custtomise a patient specific implant, that specifically conforms to the patients atypical endplate topology, thus maintaining the end outcome oof fusion, whilst allowing for a more minimal operation, when compared to standard practice for pathology with osseolysis of this degr.
Point 12: Before moving on to the conclusion section, the present reports’ limitation must be added at end of the discussion section.
Response 12: In response to point 12, liitations of this study have been included in the discussion section of the revised manuscript. I thank you again for highlighting the fact this had not been includeed on the first version of the manuscript.
Point 13: Conclusion section is missing, provide it.
Response 13: In reesponce to point 13, a conclusion has been added to the article.
Point 14: Mention further research in the conclusion section.
Response 14: This has been addressed in the new revision of the manuscript.
Point 15: Literature from the last five years should be enriched to reference. MDPI reference is strongly recommended.
Response 15: in response o point 15. There has been the addition of 5 rerferences the enrich the study, with particular care to the references being recent (published within the last 5 years).
Point 16: In the whole of the manuscript, the authors sometimes made a paragraph only consisting of one or two sentences that made the explanation not clearly understood. The authors need to extend their explanation to become a more comprehensive paragraph. In one paragraph, it is recommended to consist of at least 3 sentences with 1 sentence as the main sentence and the other sentences as supporting sentences.
Response 16: In response to point 16, the authors have adressed this and made paragraph structure alterations throughout thee new manuscript so that this issue is correct. Many thanks for this insight.
Point 17: The manuscript needs to be proofread by the authors since it has grammatical and language issues.
Response 17: In response to point 17, as there has been considerable changes, the revised manuscript has been proof read by the authors.
Point 18: After revision, provide a graphical abstract for submission.
Response 18: In response to point 18. At the end of the revised manuscript there is a copy of the graphical abstract.

Reviewer 3 Report
Thank you for your nice contribution to the literature and sharing the your experience with the application of PSI. It is encouraging to know techniques are being utilized to fit the patients own unique anatomy and not "distort to fit the implant."
Looks like a nice case with good outome.
Can the authors add information on the type of 3D printer and metal used for implant. Older generations of metal 3 D printers have had poor application to industry b/c were not durable parts.... the newer metal 3D printers I know are better and are being applied as "additive manufacturing" so must have overcome this barrier. It would be nice to know some more info in this so we can confidently tell our patients the new technology is durable.... despite the fact that the represented case showed a result of a failed implant ;)
Author Response
Dear reviewer,
I thank you for taking the time to review the manuscript and offering helpful and insightful points to improve it. As several changes needed to occur the manuscript has been revised. With respect to the point raised by yourself, the type of printer, the material used and the type of 3D printing performed has been included in the article now. Additional information on why the specific alloy was used and the reason for use of direct metal laser sintering was employed are now included in the article. spelling and grammatical improvements have also been performed.
I thank you again for your help in reviewing this article.
kind regards,
Round 2
Reviewer 2 Report
Reviewers greatly appreciate the efforts that have been made by the author to improve the quality of their articles after peer review. I reread the author's manuscript and further reviewed the changes made along with the responses from previous reviewers' comments. Unfortunately, the authors failed to make some of the substantial improvements they should have made making this article not of decent quality with biased, not cutting-edge updates on the research topic outlined. In addition, the author also failed to address the previous reviewer's comments, especially on comments number 4 (too weak, like a replication study), 5 (not comprehensive explanation), and 7 (not incorporate the suggested literature) With all due respect, the reviewer opposed this article to be published and must be rejected. Thank you very much for the opportunity to read the author's current work.
Author Response
Dear reviewer,
I appreciate the time you have taken to re-review the article and agin provide an informative way to again improve the piece of literature.
Firstly the article has been re-reviewed and grammatical errors corrected and the language used improved. Of note three are still several sentences which do incorporate the passive voice, however this is done to improve the readability of the article.
Secondly, several aspects of the article have been changed to improve the article as suggested by the reviewer. In particular, significant changes in the introduction, have been made to provide sufficient background. The cited references have all been reviewed and deemed relevant to the research. As this is a case report all authors agree that the research design depicted in this piece is appropriate. In addition the methods section, is felt by all authors to adequately address the patients clinical scenario, the process of patient specific implant design considerations and the procedure of implantation In the results section the postoperative key measures (clinical improvement and development of osteointegration), are clearly described and figures highlighting this are clear. The conclusion has also been reviewed and deemed to address and clearly support the results section.
In addressing the further comments made by the reviewer, this new version of the article has addressed the major changes the reviewer has commented on. In particular:
Comment 4 - the authors have re-written the introduction to highlight the novel aspect of 3D printing of patient specific implants in the management of the failed cervical arthroplasty (total disc replacement) with severe osteolysis. In particular although 3D printed implants have been used in spinal revision surgery (from failed fusion surgery), there are no documented cases of revisions in cases of total disc replacement failures. This case report being the first one describing this. In addition the authors have made significant changes to the introduction to highlight the fact that where normal clinical practice would normally entail a large procedure (vertebrectomy) to address this level of osteolysis, the used of 3D technology can allow for much smaller operations in this pathology. This highlighting a cutting-edge update in revision strategy for cases of total disc replacement failure with severe osteolysis.
Comment 5 - the introduction has undergone significant changes to improve the explanation of both key technological processes but also explain key paper findings. In so doing aiding the reader understand current practice but also hence the research gap in current medical literature.
Comment 7- the authors have again adjusted the introduction to included literature addressing a route to minimise the need for revision surgery. In addition the authors have included the reference that was suggested by the reviewer. These literary references being published within the last 5 years.
The authors hope that the changes made, satisfy the reviewers critique.
Kind regards,